# The Use of Polymers to Enhance Post-Orthodontic Tooth Stability

**DOI:** 10.3390/polym15010103

**Published:** 2022-12-27

**Authors:** Niswati Fathmah Rosyida, Ika Dewi Ana, Ananto Ali Alhasyimi

**Affiliations:** 1Department of Orthodontics, Faculty of Dentistry, Universitas Gadjah Mada, Yogyakarta 55281, Indonesia; 2Department of Dental Biomedical Sciences, Faculty of Dentistry, Universitas Gadjah Mada, Yogyakarta 55281, Indonesia; 3Research Collaboration Center for Biomedical Scaffolds, National Research and Innovation Agency (BRIN), Jakarta 10340, Indonesia

**Keywords:** polymer, relapse, orthodontic, retention, statin, epigallocatechin, hydrogel, bisphosphonate, car-bonated hydroxyapatite, advanced platelet-rich fibrin

## Abstract

Relapse after orthodontic treatment occurs at a rate of about 70 to 90%, and this phenomenon is an orthodontic issue that has not yet been resolved. Retention devices are one attempt at prevention, but they require a considerable amount of time. Most orthodontists continue to find it challenging to manage orthodontic relapse; therefore, additional research is required. In line with existing knowledge regarding the biological basis of relapse, biomedical engineering approaches to relapse regulation show promise. With so many possible uses in biomedical engineering, polymeric materials have long been at the forefront of the materials world. Orthodontics is an emerging field, and scientists are paying a great deal of attention to polymers because of their potential applications in this area. In recent years, the controlled release of bisphosphonate risedronate using a topically applied gelatin hydrogel has been demonstrated to be effective in reducing relapse. Simvastatin encapsulation in exosomes generated from periodontal ligament stem cells can promote simvastatin solubility and increase the inhibitory action of orthodontic relapse. Moreover, the local injection of epigallocatechin gallate-modified gelatin suppresses osteoclastogenesis and could be developed as a novel treatment method to modify tooth movement and inhibit orthodontic relapse. Furthermore, the intrasulcular administration of hydrogel carbonated hydroxyapatite-incorporated advanced platelet-rich fibrin has been shown to minimize orthodontic relapse. The objective of this review was to provide an overview of the use of polymer materials to reduce post-orthodontic relapse. We assume that bone remodeling is a crucial factor even though the exact process by which orthodontic correction is lost after retention is not fully known. Delivery of a polymer containing elements that altered osteoclast activity inhibited osteoclastogenesis and blocking orthodontic relapse. The most promising polymeric materials and their potential orthodontic uses for the prevention of orthodontic relapse are also discussed.

## 1. Introduction

Society today is experiencing an increasing interest in cosmetic dentistry, making orthodontics an essential treatment field. Orthodontic treatment has become one of the most popular treatments in cosmetic dentistry; it is used to correct malocclusion, enhance occlusion, and attain dentofacial harmony [1]. Even after several years of post-treatment stabilization, corrected teeth frequently relapse. Relapse can be explained as a phenomenon that occurs after treatment, in which the corrected tooth arrangement returns to its original pre-treatment position [2]. Relapse following orthodontic treatment occurs at a rate that ranges from around 70 to 90%, and this occurrence is an orthodontic problem that has not yet been resolved [3]. Retention is regarded as the last and most important stage in ensuring that dental components are kept in the correct place after active orthodontic tooth movement. According to clinicians, retainer should be kept in place for as long as a perfect alignment is needed [4]. Fixed bonded retainers are the most commonly used retainer, particularly for the mandibular arch, due to their stability, high effectiveness, independence of patient compliance, simplicity of installation, virtually invisibility (Figure 1), and patient acceptance [5]. These retention techniques are typically more effective for the mandibular arch, although they can also be employed for the maxillary arch. Despite the fact that this appliance does not require the patient’s active assistance, as detachable retainers do, its main drawback is the risk of detachment. Factors that contribute to this phenomenon include insufficient adhesive, setting deformities, retainer trauma, poor fatigue resistance, and adverse occlusal interactions that induce composite abrasion. It’s possible that using universal restorative adhesive after enamel pre-etching could be a dependable alternative for preventing the detachment of fixed retainers. Universal restorative adhesive contains the monomer 10-Methacryloyloxydecyl dihydrogen phosphate (10-MDP), which is capable of forming a chemical interaction with dental substrates [6]. Nevertheless, relapse is still a possibility 10 years after the retainer has been removed [7].

The process of alveolar bone remodeling was found to play a significant role in orthodontic relapse occurrence, as discovered by Franzen et al. in their animal study [8]. The process of bone remodeling can be regarded as a kind of turnover, in which newly created bone replaces older bone [9]. The dynamic process of bone remodeling is controlled by osteoclasts, which are cells that dissolve bone; osteoblasts, which are cells that produce new bone; and bone mesenchymal stem cells. All of the aforementioned cells communicate and collaborate to achieve bone remodeling [10]. Relapse can be effectively decreased by biological agents that inhibit bone resorption and stimulate bone formation [11]. These results indicate that controlling alveolar bone remodeling after active orthodontic tooth movement is a crucial method for preventing relapse.

Polymer science has been the most popular field of study due to its vast applicability in engineering modern materials for the enhancement of structural and functional qualities in clinical and biomedical applications. A variety of polymers have optimal qualities, and the chemical modification of these polymers can improve their cytocompatibility, bioactivity, and antibacterial capabilities [12,13]. Several studies have revealed that polymeric materials may be used in tissue engineering to reconstruct cartilage, bone, and heart valves, and as skin, hip, and dental implants [14,15,16,17,18]. Polymers have gained a great deal of attention from academics in recent years due to their potential use in the rapidly developing field of orthodontics [19]. The local injection of epigallocatechin gallate-modified gelatin inhibits osteoclastogenesis and has the potential to be evolved into an unique therapeutic strategy that modifies tooth movement and prevents orthodontic relapse [20]. Topically applied bisphosphonate risedronate with gelatin hydrogel reduces relapse 7 days after tooth stability in a dose-dependent manner. The proposed gelatin hydrogel method may administer risedronate to a specific area and give local effects, which is advantageous in orthodontic therapy [21]. Hydrogel carbonated hydroxyapatite-incorporated advanced platelet-rich fibrin is effective as a biological retainer for reducing orthodontic relapse. In this study, the method of applying an osteoinductive and osteoconductive substance was minimally invasive, cost-effective, and suitable for minimizing relapse after active orthodontic tooth movement [22]. In a rat model of orthodontic tooth movement, encapsulating simvastatin into exosomes generated from periodontal ligament stem cells improved simvastatin solubility and increased the inhibitory impact of relapse. Interestingly, exosomes of periodontal ligament stem cells administered locally can help prevent relapse as well [23]. The purpose of this review was to present an overview of the use of polymers as a material to lower the risk of post-orthodontic relapse.

## 2. Application of Hydrogel Carbonated Hydroxyapatite-Incorporated Advanced Platelet-Rich Fibrin Improves Post-Orthodontic Tooth Stability

Tissue engineering technologies have previously been promoted for manipulating alveolar bone remodeling, preventing orthodontic relapse, and improving tooth position stability. Because of its well-controlled calcium release and bone forming capacity, carbonate apatite (CHA) has great potential for bone tissue engineering [24]. Since it exhibits structural similarity to the interconnecting porous structure of bone, CHA is regarded as a good biomaterial for promoting alveolar bone rebuilding [25]. By elevating calcium and phosphate levels in the local area, which are essential for bone development, CHA promotes bone remodeling. The activity of the osteoblasts is controlled by the release of calcium and phosphate ions into the surrounding tissue. High levels of extracellular calcium also inhibit osteoclastic development and promote DNA synthesis and chemotaxis in osteoblastic cells [22]. CHA has also gained prominence for its capacity to function as a medication delivery system for protein transport into living cells [26].

A further advantage of CHA is its capacity to function as a drug delivery system in controlled release innovation [27]. One of the most recent issues in tissue engineering is the advancement of controlled release methods for bone tissue augmentation. The controlled release system is seen to be promising because it can convert materials with a low molecular weight into a system with a higher molecular weight, preventing degradation before the medicine begins to operate. In order to manage the water content of the hydrogel system, gelatin hydrogel was selected for this study to provide controlled release and degradability using a cross-linking technique. The hydrogel can be degraded enzymatically to produce water-soluble gelatin fragments, allowing bioactive-loaded components to be released [28].

Growth factors (GFs) are natural polypeptides that stimulate extracellular matrix synthesis and enhance osteoblast development [29]. Taking into account the presence of GFs, it is hypothesized that a suitable incorporation of hydrogel CHA exhibiting controlled release and GF could achieve more favorable bone regeneration outcomes. Platelet-rich fibrin (PRF) is a new generation of GF-rich platelet concentrate that is more beneficial than other platelet concentrates, such as platelet-rich plasma (PRP), due to its simple preparation, low cost, and absence of anticoagulants, such as bovine thrombin and calcium chloride, for platelet activation [30]. Endogenous thrombin, released during centrifugation, can quickly activate PRF. The restriction on the usual use of bovine thrombin due to the high risk of coagulopathy from antibody development [31] has limited its use. The use of calcium chloride and thrombin to coagulate platelets into a gel and engage the contained GF initiates a burst release and activation of all the GFs of PRP simultaneously; thus, the period of action of PRP is brief. Meanwhile, PRF can maintain GF activity for a substantially longer duration and efficiently induce bone repair., Kobayashi et al. [32] revealed that over a period of 10 days, PRF released GFs continuously and consistently. Another technique to improve PRP’s half-life limits is to use a biodegradable gelatin hydrogel drug delivery technology. The release of growth factors to ischemic areas is controlled by biodegradable gelatin hydrogel. Platelets release growth factors when impregnated into biodegradable gelatin hydrogel [33]. Gelatin molecules electrically and physiochemically immobilize growth factors in the hydrogel [34]. After injecting PRP and biodegradable gelatin hydrogel into ischemic tissue, the growth factor-impregnated hydrogel slowly releases growth factors into the tissue over 2 weeks, resulting in more successful angiogenesis [35]. Specifically, angiogenesis may enhance oxygen and nutrient supply and offer a pathway for bone precursor cells to reach the intended area [29].

Advanced PRF, also known as aPRF, is a novel form of PRF modification that is made by slowing down the centrifugation speed of the conventional methods of fibrin preparation and increasing the amount of time that the centrifugation process requires. Platelet concentrations are increased as a result of this modification because during the centrifugation process, fewer cells settle to the bottom of the tubes, and a greater quantity of proteins, including platelets, are left in the higher part of the tubes, where the clot is isolated. This results in higher platelet concentrations [32,36]. The platelet count in PRF releasate was found to be 2.69 times greater than the count of platelets in whole blood, according to Burnouf et al. [37], whereas Alhasyimi et al. [38] demonstrated that the platelet count in aPRF releasate was 4.78 times higher than the count of platelets in whole blood.

The considerable increases in alkaline phosphatase activity at days 7 and 14 after orthodontic debonding suggest that intrasulcular injection of controlled release hydrogel CHA, including aPRF, has the potential to promote alveolar bone remodeling and prevent orthodontic relapse [25]. ALP has been identified as a measure of osteoblastic activity during bone formation [39]. The study found a link between alveolar bone remodeling and variations in ALP activity present in GCF. It suggested that osteoblastic cell activity was boosted when ALP levels were significantly elevated [40]. The increased osteoblast activity indicates that new bone is forming. Osteoblasts must continually drive bone regeneration to prevent relapse [22]. Osteoblasts subsequently commence bone apposition by generating fresh bone matrices within the osteoclast-formed trenches and tunnels on the bone’s surface [41].

Hydrogel CHA with aPRF was shown to be effective in preventing orthodontic relapse in a rabbit model following tooth movement. CHA−aPRF intrasulcular injection has the potential to minimize orthodontic relapse by stimulating osteoprotegerin (OPG) expression and inhibiting the receptor activator of the nuclear factor−κB ligand (RANKL) level. Inhibition of osteoclastogenesis and osteoclast activity by local injection of CHA−aPRF could improve orthodontic retention, according to the findings of this study [38]. OPG is a natural receptor expressed by osteoblasts which inhibits osteoclast differentiation and activity by binding to the RANKL and blocking RANKL from interacting with RANK. The binding of RANKL and the RANK receptor leads to rapid differentiation of hematopoietic osteoclast precursors to mature osteoclasts [42]. During orthodontic relapse in rabbits, a similar study found that injections of CHA and CHA hydrogel−aPRF favorably upregulated transforming growth factor−β1 (TGF−β1) and bone morphogenic protein−2 (BMP−2) expression, but did not increase Runt−related transcription factor−2 (Runx−2) levels [43]. The expression of TGF−β1 and BMP−2 is crucial to the process of osteoblastogenesis. It is thought that osteoblastogenesis can balance out the activity of osteoclasts [44]. Runx−2 expression is reinforced and mesenchymal stem cell development is encouraged by a signaling cascade started by TGF−1 and BMP−2 [45]. TGF−β1 promotes osteoblast proliferation by recruiting osteoblast precursors or matrix-producing osteoblasts via chemotactic attraction and attempting to prevent osteoblast apoptosis [46]. TGF−β1 is the most potent bone formation stimulator, increasing fibroblast proliferation and stimulating collagen synthesis [47]. Furthermore, increased BMP−2 expression can induce osteoblast maturation and initiate alveolar bone formation to effectively prevent relapse [48]. BMP−2 can increase bone mass by decreasing osteoclastogenesis activity via the RANKL−OPG pathway [49]. Osteoclastogenesis inhibition has been shown to be effective in lowering relapse rates following orthodontic tooth movement [50]. Meanwhile, Runx−2 regulates the expression of RANKL and OPG by stimulating osteoclast differentiation [51]. However, the molecular mechanisms by which Runx−2 impedes improvements in osteoclastogenesis require additional study [43]. Figure 2 summarizes the process by which hydrogel CHA−aPRF prevents orthodontic relapse.

## 3. Statins’ Inhibitory Effect on Relapse after Orthodontic Treatment

The statin family of drugs is an effective treatment for arteriosclerotic cardiovascular disease. Statins have the capability to inhibit 3-hydroxy-3-methyl glutaryl reductase, a rate-limiting enzyme in the cholesterol biosynthesis mevalonate pathway [52]. Statins have been shown to have numerous favorable effects on human health, including anabolic effects on bone metabolism in various ways, in addition to their cholesterol-lowering properties. They promote the osteoblastic differentiation of bone marrow stem cells by upregulating BMP-2 gene expression and angiogenesis. Statins may also promote bone formation by preventing osteoblast apoptosis [53,54,55]. Statins suppress osteoclastic bone activity during periods of high bone turnover, resulting in the reduction of bone resorption. This effect involves the modulation of the receptor activator of nuclear kappa B (RANK), RANKL, and OPG, ultimately suppressing osteoclastogenesis [56,57]. Thus, their ability to stimulate bone formation while also exhibiting pleiotropic effects, such as anti-inflammatory and immunomodulatory properties, could justify their use in orthodontic relapse prevention [58]. Because osteoclastic resorption and osteoblastic formation of surrounding alveolar bone are important factors in relapse, stimulating alveolar bone formation or inhibiting bone resorption after orthodontic tooth movement should prevent relapse. Figure 3 illustrates the statin’s role in reducing orthodontic relapse. Given the bone modulation properties of statins, their possible effect of blocking orthodontic relapse could be a consideration in orthodontic treatments. Table 1 summarizes the data from an animal study on the efficacy of prescribed statins in preventing postorthodontic relapse.

On the basis of their solubility, Statins are classified as lipophilic. Due to its low therapeutic potential and poor permeability, the distribution of a lipophilic substance presents a significant challenge for conventional delivery systems. According to stud-ies, nanoscale-sized preparations can increase drug permeability by disturbing the li-pid layer and lengthening drug retention duration at the site of action [58,59]. Because it has stronger thermodynamic stability and drug solubilization capability than emul-sion and other dispersion systems, nanoemulsion may be a potential carrier delivery strategy for hydrophobic drugs. It also has a longer shelf life and requires little external energy to manufacture. A nanoemulsion is a dispersed system formed of na-noscale-sized (20–200 nm diameter) droplets of a solvent composed of an oil phase and a water phase and stabilized by the appropriate surfactant [60]. Statins in the form of nanoemulsions appear to be promising for orthodontic applications.

The statins used in the 7 studies listed above are atorvastatin and simvastatin. Differences in the chemical structures of the two statins, the efficacy of one drug over the other, and dose dependence are all factors that may contribute to inconsistencies in the conclusions and their extrapolation to human subjects. This complicates the already difficult task of predicting the future outcomes of statin administration in humans. Simvastatin is soluble in lipids, which may be one reason why its anti-relapse effect was not particularly remarkable. In a novel study, exosomes derived from periodontal ligament stem cells (PDLSCs-Exo) were used as drug carriers to load simvastatin into exosomes. This was accomplished using ultrasound and co-incubation. Simvastatin can be loaded and rendered more soluble by PDLSCs-Exo. Interestingly, during orthodontic relapse, PDLSCs-Exo may control local alveolar bone remodeling by carrying various osteogenesis signaling molecules. As a result, even when PDLSCs-Exo is injected alone, it can still prevent relapse after OTM [23].

## 4. Epigallocatechin Gallate-Modified Gelatin (EGCG-GL) Inhibits Bone Resorption and Tooth Movement in Rats

Downstream of RANKL is an intracellular signaling molecule called reactive oxygen species (ROS) [66]. As a result, scavenging ROS is an attractive strategy for inhibiting osteoclasts. Inhibition of osteoclastogenesis can be achieved by activating nuclear factor E2-related factor 2 (Nrf2) [67,68,69]. Epigallocatechin gallate’s (EGCG) ability to stimulate Nrf2-mediated anti-oxidation and ROS scavenging slows down orthodontic tooth movement. Nevertheless, EGCG injections must be repeated if they are to successfully slow OTM by blocking osteoclastogenesis [70]. A previous study observed that repetitive local injections of EGCG solution reduced osteoclastogenesis and as a result, slowed orthodontic tooth movement [67]. However, frequent local injections are not a viable therapeutic option for orthodontics. This issue was addressed by developing EGCG-GL, since it would benefit orthodontic patients by increasing anchorage strength and decreasing the rate of OTM [20]. In 2018, the first vacuum-heated EGCG-modified gelatin sponges for bone regeneration therapy were established. The observed increase in bone formation after vacuum heating can be attributed in part to the effect of the reduced degradability of the sponge caused by DHT cross-linking, which offers a scaffold for cells. The results indicate that the pharmacological impact of EGCG survives vacuum heating and is associated with an increase in bone formation [71]. EGCG-GL was produced by chemically cross-linking EGCG and gelatin using a simple and eco-friendly synthetic approach [70], while preserving EGCG’s activity [71]. Mixing EGCG-GL with bromelain, a combination of proteolytic enzymes isolated from pineapples, maintains the steady release of EGCG by gradually breaking down the gelatin [20].

EGCG inhibits LPS-induced RANKL expression in osteoblasts [72]. Furthermore, EGCG enhances the prostaglandin-stimulated production of OPG in osteoblasts in a synergistic manner [73,74]. Consequently, EGCG reduces the RANKL/OPG ratio at the location, which indirectly suppresses osteoclastic differentiation. Improvements in orthodontic retention may be possible through the suppression of osteoclastogenesis and osteoclast activity [38]. Using flow cytometry, EGCG-GL showed inhibited RANKL-mediated intracellular ROS generation in RAW 264.7 cells. These findings indicate that EGCG-GL inhibits RANKL signaling through intracellular ROS formation [20].

## 5. The Potential Benefits of Using Bisphosphonate Risedronate Hydrogel to Prevent Orthodontic Relapse Movement

Bisphosphonates are medications used to treat diseases of the bone metabolism, such as osteoporosis. Bisphosphonates bind tightly to hydroxyapatite and inhibit bone resorption. They specifically target calcified tissues, where they are absorbed selectively by bone-resorbing osteoclasts [75]. Once internalized, bisphosphonates downregulate the ability of osteoclasts to resorb bone by interfering with cytoskeletal organization and the formation of the ruffled border, resulting in apoptotic cell death [76,77]. Bisphosphonates have been proposed in orthodontics as a possible means of controlling relapse and even generating “pharmacological anchorage”. The clinical utility of bisphosphonates stems from their capacity to prevent bone resorption. Anchorage loss and post-treatment relapse are two major concerns in orthodontic treatment [78]. Bisphosphonates were found to inhibit tooth movement in rats by decreasing osteoclast formation. Bisphosphonates also helped to prevent root resorption caused by orthodontic tooth movement. These findings imply that bisphosphonate may be beneficial for regulating orthodontic tooth movement and as a potential inhibitor of root resorption during orthodontic tooth movement and relapse after orthodontic tooth movement [79].

Bisphosphonates may cause bisphosphonate-related jaw osteonecrosis, an oral necrotic bone condition [80]. The duration, dosage, and intravenous and oral administration of bisphosphonate can cause a systemic effect [81]. All previously cited studies utilized pure bisphosphonates, without a carrier, to successfully treat periodontal disease. Adachi et al. revealed that the local injection of risedronate effectively lessens relapse, but with systemic side effects, including an increase in tibial bone mineral density [82]. Given that bisphosphonates affect the entire skeleton, the most effective method for treating periodontal bone loss would be a topical application [83]. On days 14 and 21 after active orthodontic tooth movement, the intrasulcular administration of bisphosphonate risedronate hydrogel altered the osteoclast-to-osteoblast ratio and raised alkaline phosphatase levels, and 7 days after the tooth stabilization period, the application of bisphosphonate risedronate with gelatin hydrogel efficiently reduces relapse in a dose-dependent manner. Hydrogel risedronate bisphosphonate improved the proliferation and maturation of osteoblasts, which play a crucial role in bone production, hence enhancing tooth stability during orthodontic movement. In orthodontics, the developed gelatin hydrogel technology can be used to administer risedronate precisely where it is needed for localized effects. These findings demonstrate the significance of bisphosphonate risedronate hydrogel in the bone remodeling process; hence, it has the ability to prevent relapse [21,84]. Figure 4 depicts the role that bisphosphonates play in reducing orthodontic relapse.

The standard of treatment in the medical field is shifting toward more minimally invasive techniques. The practice of dentistry known as "minimally intrusive dentistry" adheres to a philosophy that emphasizes the integration of prevention, remineralization, and minimal intervention in the placement and repair of restorations. The goal of treatment can be accomplished with minimally invasive dentistry by employing the least invasive surgical technique, removing the smallest possible amount of healthy tissue, lowering the risk of soft tissue reorganization, and focusing solely on the factors that pose the greatest potential for complications [84,85]. The use of a topical treatment is one example of a procedure that is regarded to be minimally invasive.Utari et al. employed a topical hydrogel risedronate formulation to minimize relapse in guinea pigs; however, its placement into the gingival sulcus was still difficult [86]. When compared to pure bisphosphonate solution, the risedronate emulgel using virgin coconut oil exhibited a regulated medication release, and it may be administered topically to prevent relapse [87]. Emulgel is an oil-in-water or water-in-oil emulsion that has been combined with a gelling agent. Emulgel preparations exhibit a number of advantages, including hydrophobic drug properties and a high loading capacity, which allows for ease of production, inexpensive production costs, and controlled drug delivery [88].The hydrophobic stratum corneum, which acts as a barrier to stop medication permeability, is the most significant possible issue associated with topical distribution. Delivering hydrophilic or larger molecular weight medicinal medicines across crystalline barriers thus becomes difficult. When compared to alternative carriers like microemulsions, liposomes, or solid lipid nanoparticles, nano emulsion may offer a number of major advantages including minimal irritancy, strong penetration ability, and high drug-loading capacity for topical delivery [89,90].

## 6. Conclusions

After receiving orthodontic treatment, retention is one of the most important methods that can be used to prevent orthodontic relapse. In spite of this, the mechanism of orthodontic relapse is still unknown, despite the fact that relapse is frequently observed in some patients regardless of the effective use of a retainer. Although the precise mechanism by which orthodontic correction is lost after retention is not fully understood, we presume that bone remodeling is a major contributor. Inhibition of osteoclastogenesis and a delay in orthodontic tooth movement were observed after the delivery of a polymer containing materials that influenced osteoclast activity. Osteoclastogenesis is strongly linked to relapse. PDL space widens a few days after relapse movement, which coincides with the appearance of the first osteoclast progenitor cells at the compression sites in the alveolar crest vasculature and marrow spaces. When compared to the sites of tension, compression tends to have a greater number of osteoclasts present. During tooth movement, proinflammatory cytokines are also produced, which points to the significance of inflammation in the process of initiating osteoclastogenesis. Compressive forces trigger a response from the tissue biomarker RANKL. On the other hand, an increase in the osteoprotegerin biomarker leads to a decrease in RANKL, which in turn prevents tooth movement. This new finding can serve as a key strategy for developing materials that effectively and efficiently prevent orthodontic relapse.

Numerous novel techniques exist for inhibiting osteoclastogenesis and preventing orthodontic relapse, but local application utilizing a drug delivery system is likely the most novel and well-controllable technique, as it provides the most effective control. Controlled release will be the result of the interaction between the drug and the polymer. Polymers are used to improve drug stability and facilitate release. For instance, CHA hydrogel was developed as a drug delivery system because of its capacity to perform the role of an intracellular protein transporter. The hydrogel has the ability to preserve the three-dimensional structure of proteins, such as the growth factors in aPRF, while they are being transported, thereby preventing the proteins from becoming denatured or degraded before they reach the intended site. Moreover, emulgel possesses a mucoadhesive drug delivery system, which interacts with the mucus layer on the surface of the mucosal epithelium and mucin molecules. This interaction occurs by forming intensive contact between the drug and the target area. Consequently, the retention time of drug preparations in the intended application is lengthened.

The review presented here analyzes the prospects for the use of polymeric materials in the field of dentistry, particularly in orthodontics. The improvements detailed in this study chart a new course for relapse prevention materials, with the goal of increasing patients’ quality of life. Despite the fact that the overall clarity of the evidence limits prospective recommendations for human trials, the outcomes of this analysis suggest a direction for further research. Furthermore, it is critical that any future animal research follows defined protocols. These protocols should consider the reproduction of human clinical circumstances in terms of the timing, dose equivalence, and route of medication administration, as well as the peculiarities of the mechanisms that cause tooth movement and the methods used to evaluate relapse. It is critical to calculate the correct sample size in order to increase the dependability of the findings and the overall impact of the research.

## Figures and Tables

**Figure 1 polymers-15-00103-f001:**
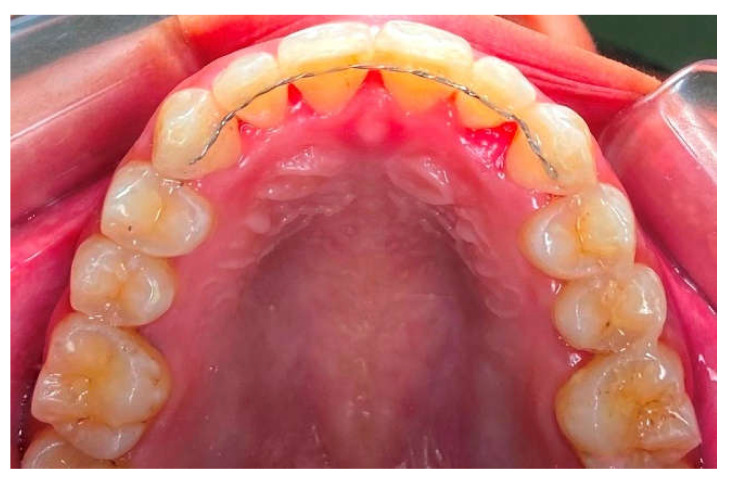
Fixed bonded retainer.

**Figure 2 polymers-15-00103-f002:**
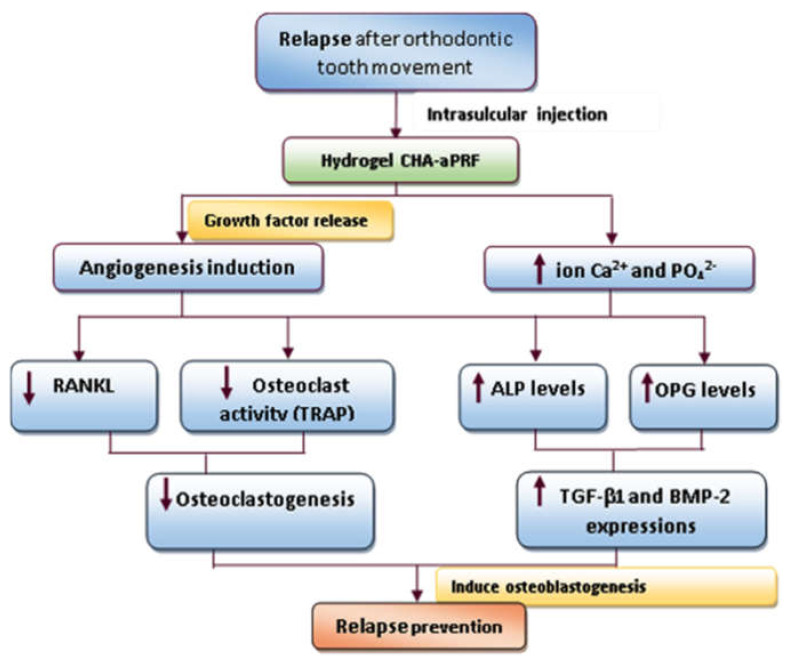
The mechanism of how hydrogel CHA−aPRF works to prevent orthodontic relapse.

**Figure 3 polymers-15-00103-f003:**
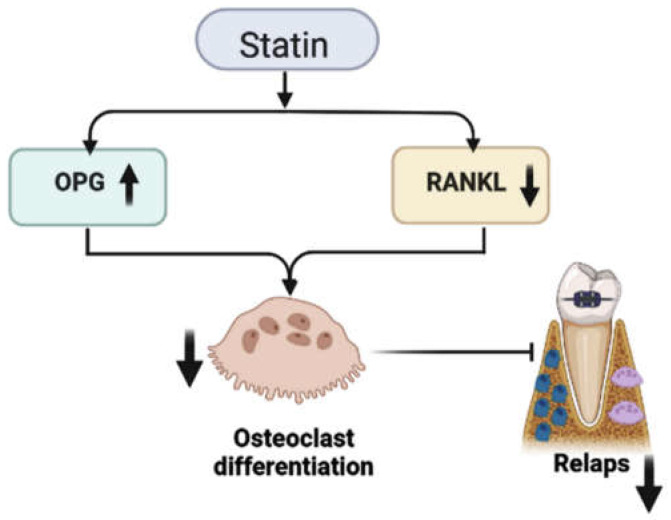
Statin’s method of action for preventing orthodontic relapse.

**Figure 4 polymers-15-00103-f004:**
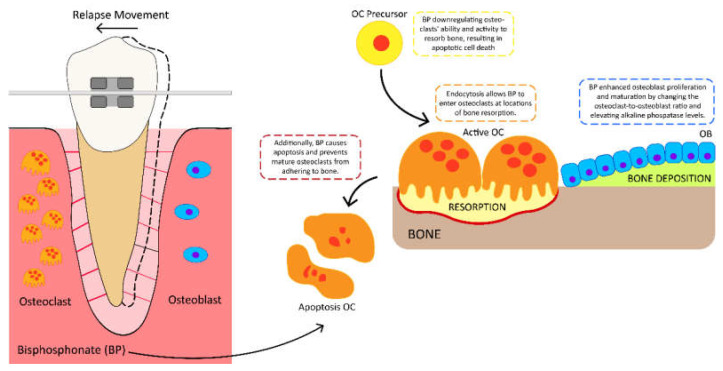
The mechanism of action of bisphosphonate (BP) in the prevention of orthodontic relapse. BP can enter the osteoclasts at sites of bone resorption via endocytosis. BP inhibits the capability and activity of osteoclasts, which causes apoptotic cell death. BP also restricts mature osteoclasts from attaching to bone. By altering the osteoclast-to-osteoblast ratio and raising alkaline phosphatase levels, BP promotes osteoblast proliferation and maturation. BP: bisphosphonates; OC: osteoclast; OB: osteoblast.

**Table 1 polymers-15-00103-t001:** The effect of statins in animal models of orthodontic relapse.

Author(s)	Administration’s Route, Type, and Dose of Statin	Result
Chen et al. [48]	SIM systemic administration 2.5 mg/kg/day, 5.0 mg/kg/day, and 10.0 mg/kg/day	Systemic administration of SIM could reduce the incidence of orthodontic relapse in rats, and a lower dose of simvastatin appeared to be more effective.
Han et al. [56]	Intraperitoneal injections of SIM, 2.5 mg/kg/day	• The SIM group showed shorter relapse distances than did the control group (*p* < 0.01). • The percentage of relapses in the test group was significantly smaller than that in the control group (*p* < 0.001).
AlSwafeeri et al. [61]	Local injection (intraligamentous and submucosal) of SIM, 0.5 mg/480μL	Local SIM administration helps postorthodontic relapse-related bone remodeling by reducing active bone resorption and increasing bone formation, but does not significantly reduce postorthodontic relapse.
Dolci et al. [62]	ATO systemic administration, 15 mg/kg	Statins reduce orthodontic relapse in rats by modulating bone remodeling. Decreased osteoclastogenesis and increased OPG protein expression explain this effect.
Vieira et al. [63]	Oral gavage of SIM, 5 mg/kg/day	SIM did not prevent relapse movement in rats, and there was no link between bone density and orthodontic relapse.
Feizbakhsh et al. [64]	Local injection of 0.5 mg/kg SIM in 1 mL solution	SIM local injection can reduce the rate of tooth movement and root resorption in dogs, but the differences were not statistically significant.
MirHashemi et al. [65]	Daily gavage of ATO, 5 mg/kg	In rats, ATO appeared to reduce tooth movement; however, its effect on osteoclasts, particularly regarding osteoclastic activity, requires additional research.

SIM: simvastatin; ATO: atorvastatin.

## Data Availability

Not applicable.

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
