# Peer review of "The Use of Polymers to Enhance Post-Orthodontic Tooth Stability"

_polymers, 2022, doi:10.3390/polym15010103_

Round 1

Reviewer 1 Report

To Authors: 

The paper has been well researched and written with great insights in the field of use of hydrogels in orthodontics. Congratulations to all the authors. 

However, since this is a review article, it is expected that you describe some of your own research and elucidate some novel findings in the field from your own recent studies in the same field with reference to work done by other researchers. There is very less new findings in this paper. 

Also, this review article focuses on orthodontic aspects of the work, like tissue engineering, but very brief explanation of polymer engineering, physics or chemistry. I think this article is beyond the scope of this journal 'Polymers'. But can be considered for other MDPI journal more suitable for tissue engineering or biomedical study. 

Author Response

Dear

Editor-in-Chief and Reviewer(s), Polymers

Thank you for giving us this opportunity; it is an honor. We have been working throughout the review. The following summarizes our overall thoughts on the review and some changes we made in response to reviewer feedback.

Referee: 1 

Comments to the Author 

The paper has been well researched and written with great insights in the field of use of hydrogels in orthodontics. Congratulations to all the authors. 

However, since this is a review article, it is expected that you describe some of your own research and elucidate some novel findings in the field from your own recent studies in the same field with reference to work done by other researchers. There is very less new findings in this paper. 

Also, this review article focuses on orthodontic aspects of the work, like tissue engineering, but very brief explanation of polymer engineering, physics or chemistry. I think this article is beyond the scope of this journal 'Polymers'. But can be considered for other MDPI journal more suitable for tissue engineering or biomedical study. 

Author's Reply to the Review Report

Thank you for your punctiliously review. We have followed the suggestion and revised the paper based on reviewer’s comments. We have already added some novel findings in the present manuscript as well, particularly in conclusion (yellow highlighted). Furthermore, Biomacromolecules, Biobased, and Biodegradable Polymers is one section of the "Polymers" journal that, in our opinion, relate to our areas of the work. Thank you so much

We really hope the manuscript will receive your kind consideration and the manuscript will be acceptable.

Sincerely,

*******

Reviewer 2 Report

Manuscript of considerable interest for  the dental sector, in particular for orthodontics , before evaluating its possibility of publication , it needs a major revision .
1. ABSTRACT: further highlighting of statistically significant data.

2. KEYWORDS: few.

3. INTRODUCTION: the other principles of pre boding prophylaxis already studied by the research group of prof .V. lanterl and cossellu.
10.4103/2278-0203.197392
Pre-bonding prophylaxis and brackets detachment: An experimental comparison of different methods.
International Journal of Clinical Dentistry
Volume 7, Issue 2, Pages 191 - 1971 January 2014

4.Very confused results highlight staticly significant data so the reader can determine the differences at a glance.

5. Materials and methords:: well described

6.DISCUSSION: add the minimal invasive approach as future goals,  the use of all the minimal invasive systems already studied by the research group of prof.Scribante.
10.3390/microorganisms10040675

7.CONCLUSION,:add a proactive action through natural substances.

Author Response

Dear

Editor-in-Chief and Reviewer(s), Polymers

Thank you for giving us this opportunity; it is an honor. We have been working throughout the review. The following summarizes our overall thoughts on the review and some changes we made in response to reviewer feedback.

Referee: 2

Comments to the Author 

Manuscript of considerable interest for the dental sector, in particular for orthodontics, before evaluating its possibility of publication, it needs a major revision .

  1. ABSTRACT: further highlighting of statistically significant data.

  1. KEYWORDS: few.

  1. INTRODUCTION: the other principles of pre boding prophylaxis already studied by the research group of prof .V. lanterl and cossellu.

10.4103/2278-0203.197392

Pre-bonding prophylaxis and brackets detachment: An experimental comparison of different methods.

International Journal of Clinical Dentistry

Volume 7, Issue 2, Pages 191 - 1971 January 2014

4.Very confused results highlight staticly significant data so the reader can determine the differences at a glance.

  1. Materials and methords:: well described

6.DISCUSSION: add the minimal invasive approach as future goals, the use of all the minimal invasive systems already studied by the research group of prof.Scribante.

10.3390/microorganisms10040675

7.CONCLUSION,:add a proactive action through natural substances.

Author's Reply to the Review Report:

Thank you for the fine suggestion We have already changes based on reviewer comments (highlighted)

  1. ABSTRACT: further highlighting of statistically significant data.

Thank you for the fine suggestion. Since it is a review paper, we did not make some statistical analysis, further, we highlight some of novel finding resumed from the review in abstract (highlighted)

  1. KEYWORDS: few.

I appreciate the good advice you provided. Based on reviewer suggestions, we have already changed and added the keywords (highlighted)

  1. INTRODUCTION: the other principles of pre boding prophylaxis already studied by the research group of prof .V. lanterl and cossellu. 10.4103/2278-0203.197392. Pre-bonding prophylaxis and brackets detachment: An experimental comparison of different methods. International Journal of Clinical Dentistry Volume 7, Issue 2, Pages 191 - 1971 January 2014. I am grateful for the sound counsel that you have given. But this topic is not relevant with the polymers we used to blocking Orthodontic relapse.
  2. Very confused results highlight staticly significant data so the reader can determine the differences at a glance. Since it is a review paper, we did not make some statistical analysis but we summarized the findings in the conclusions. Thank you
  3. Materials and methords:: well described. Thank you.
  4. DISCUSSION: add the minimal invasive approach as future goals, the use of all the minimal invasive systems already studied by the research group of prof.Scribante. 10.3390/microorganisms10040675. I appreciate the good advice you provided. Based on reviewer suggestions, we have already changed and added this research (highlighted)
  5. CONCLUSION,:add a proactive action through natural substances.

I appreciate the good advice you provided. Based on reviewer suggestions, we have already changed and added novel finding in conclusion (highlighted)

We really hope the manuscript will receive your kind consideration and the manuscript will be acceptable.

Sincerely,

*******

Reviewer 3 Report

Dear Authors,

thank you for your work.

I think that it is valuable and deserves to be considered for publication in the journal.

In order to improve its quality I think that it is important to improve the clinical aspects of orthodontic retention.

For example you could discuss the fact the adhesion of orthodontic reteainer is influenced by the adhesive protocol considered. You could refer for example to the following

Sfondrini et al. Universal Adhesive for Fixed Retainer Bonding: In Vitro Evaluation and Randomized Clinical Trial. Materials (Basel). 2021 Mar 10;14(6):1341.

The Reviewer

Author Response

Dear

Editor-in-Chief and Reviewer(s), Polymers

Thank you for giving us this opportunity; it is an honor. We have been working throughout the review. The following summarizes our overall thoughts on the review and some changes we made in response to reviewer feedback.

Referee: 3

Comments to the Author 

Dear Authors,

thank you for your work.

I think that it is valuable and deserves to be considered for publication in the journal.

In order to improve its quality I think that it is important to improve the clinical aspects of orthodontic retention.

For example you could discuss the fact the adhesion of orthodontic reteainer is influenced by the adhesive protocol considered. You could refer for example to the following

Sfondrini et al. Universal Adhesive for Fixed Retainer Bonding: In Vitro Evaluation and Randomized Clinical Trial. Materials (Basel). 2021 Mar 10;14(6):1341. 

The Reviewer

Author's Reply to the Review Report

I am grateful to you for the excellent suggestions. We have already made adjustments in response to the comments made by reviewers (highlighted). We include the results of Sfondrini et al. Universal Adhesive for Fixed Retainer Bonding: In Vitro Evaluation and Randomized Clinical Trial. Materials (Basel). 2021 Mar 10;14(6):1341. , in the introduction

We really hope the manuscript will receive your kind consideration and the manuscript will be acceptable.

Sincerely,

*******

Round 2

Reviewer 2 Report

THE MANUSCRIPT HAS BEEN SUCCESSFULLY REVIEWED, IT CAN BE PUBLISHED

Reviewer 3 Report

Dear Authors,

thank you for your work and for the improvements you have done to the manuscript.

I think that all my comments have been addressed and that the manuscript now deserves to be published.

Good work

Yours faithfully 

The Reviewer